# The Association of Cardiometabolic Risk Factors in Parent–Child Dyads in Guam: Pacific Islands Cohort on Cardiometabolic Health Study

**DOI:** 10.3390/ijerph22040611

**Published:** 2025-04-14

**Authors:** Tanisha F. Aflague, Grazyna Badowski, Karen Mae A. Bacalia, Jaelene Renae Manibusan, Regina-Mae Dominguez, Kathryn Wood, Margaret Hattori-Uchima, Rachael T. Leon Guerrero

**Affiliations:** 1College of Natural and Applied Sciences, University of Guam, Mangilao, GU 96923, USA; gbadowski@triton.uog.edu (G.B.); rachaeltlg@triton.uog.edu (R.T.L.G.); 2Department of Nutritional Sciences, Rutgers, The State University of New Jersey, New Brunswick, NJ 08901, USA; kb1045@sebs.rutgers.edu; 3School of Health, University of Guam, Mangilao, GU 96923, USA; kwood@triton.uog.edu (K.W.);

**Keywords:** metabolic syndrome, type 2 diabetes mellitus, Pacific Islanders

## Abstract

The Western Pacific region, including Guam, has the highest prevalence of prediabetes and type 2 diabetes mellitus, which are associated with metabolic syndrome (MetS)—a cluster of preventable risk factors. Children with parents with MetS are likely to develop MetS in the future. MetS prevalence in Guam and the impact of MetS on children are unknown. Data from the Pacific Islands Cohort on Cardiometabolic Health (PICCAH) study in Guam were analyzed to determine MetS in adults and MetS risk in children using the International Diabetes Federation criteria and sex- and age-specific waist circumference values for abdominal obesity, respectively. MetS Z-scores were calculated. MetS or MetS risk indicators, including MetS Z-scores, were examined by lifestyle risk factors (parent and child: physical activity and sleep; parent only: sedentary behavior and stress; child only: screen time). The relationship between adult MetS Z-scores and child MetS Z-scores was evaluated using linear-regression analyses. Child–parent risk for MetS was directly correlated in this population. The high prevalence of adult MetS in Guam demonstrates a critical need for interventions involving both parents and children. Expanding the analysis to assess the relationships between other lifestyle factors, like diet, in parent–child dyads is necessary to refine such intervention programs.

## 1. Introduction

The prevalence of metabolic syndrome (MetS) has shown a concerning upward trend globally [1,2]. Metabolic syndrome (MetS) is a cluster of risk factors associated with an increased risk of developing cardiovascular disease (CVD) and other health problems, such as type 2 diabetes mellitus (T2DM) and stroke [3,4]. According to the National Health and Nutrition Examination Survey (NHANES) data from 1988 through 2012, the prevalence of MetS among U.S. adults increased from 25.3% to 34.2% from 1988–1994 to 2007–2012 [5]. NHANES does not collect data in Hawai’i, Alaska, and U.S. Territories due to the mobile examination centers designed to reach only the contiguous 48 states [6]. For this reason, the prevalence of MetS among Pacific Islanders is not well documented.

In 2021, the International Diabetes Federation (IDF) reported that the Western Pacific region had the highest prevalence of prediabetes and type 2 diabetes mellitus (T2DM) globally, indicative of elevated blood glucose levels [7]. In Guam, an unincorporated U.S. territory in the Western Pacific region, there were 11.7% of adults diagnosed with diabetes mellitus (DM), and the prevalence of overweight or obesity (OWOB) was 67.7% in 2019, values which were both higher than the U.S. overall 10.8% (median) and 67.0%, respectively [8]. Despite limited data, OWOB prevalence was also high among youth in Guam, affecting 39.3% of 4–19-year-olds during 2010–2014 [9]. In a more recent study, 33.8% of young children ages 2–8 years had abdominal obesity and were OWOB [9,10]. These trends underscore the need for further research into MetS prevalence in this population.

The IDF defines MetS in adults if there is high WC and any two of the following four factors are present: (1) elevated triglyceride levels (≥150 mg/dL) or the individual receiving treatment for dyslipidemia; (2) low HDL cholesterol levels (HDL < 40 mg/dL for males and females HDL < 50 mg/dL) or the individual receiving treatment for dyslipidemia; (3) high blood pressure (systolic ≥130 mmHg or diastolic ≥ 85 mmHg) or receiving treatment for hypertension; and (4) elevated fasting blood glucose levels (≥100 mg/dL) or the presence of T2DM. In children, the IDF defines MetS by age group (i.e., 6 to <10 years, 10 to <16 years, and 16 years or older) to account for age-related differences [11]. The criteria for diagnosing MetS for children in the 6 to <10 years age group have not yet been established. However, children in this age group are considered to be at risk for MetS if they have a WC above the 90th percentile for the waist-for-age ratio [11]. For children in the 10 to <16 years age group, MetS is diagnosed using the IDF cut-off points, where abdominal obesity must be present along with two or more clinical features, including elevated triglycerides, low HDL or high-density lipoprotein cholesterol, high blood pressure, or increased plasma glucose. Children ages 16 years or older follow the same IDF criteria as adults [11].

Modifiable lifestyle risk factors that are associated with one or more MetS indicators include increasing physical activity, reducing sedentary behavior, maintaining adequate sleep, managing stress, and adhering to a healthy diet [12,13]. A recent study conducted in Guam discovered that the majority of children, ages 2–8 years, did not meet the recommended guidelines for sleep duration (59.6%), sedentary screen time (83.11%), fruit (58.7%) and vegetable (99.1%) intake, or consumed sugar-sweetened beverages (SSBs) (73.7%) [10]. Furthermore, the few diet studies in Guam revealed greater SSB intake among OWOB children and adults when compared to individuals with a healthy weight [10,14,15]. SSBs have been correlated with the presence of MetS in cross-sectional studies [16]. Another dietary factor strongly associated with dyslipidemia is a diet high in saturated fat and low in monounsaturated and polyunsaturated fatty acids (MUFAs and PUFAs) [17]. Children and adults in Guam are at risk for dyslipidemia and, in combination with a high BMI, MetS related to the regular intake of foods high in saturated fat and low in MUFAs and PUFAs [9,10,12,14]. Other risk factors for MetS include socioeconomic status (SES) and parent–child relationship or metabolic inheritance [17,18,19,20,21]. Studies have shown that MetS prevalence is higher in those with a low SES [20,22]. Food insecurity data are extremely limited in Guam, yet this condition was determined to be prevalent among 51.5% of child participants who reported that food money ran out at the end of the month [10]. Studies performed in the U.S. and other countries have shown that children with at least one parent who has MetS have an increased risk of developing MetS, and the risk increases if both parents are affected [22,23,24].

Documenting the burden among adults and understanding modifiable risk factors among both children and adults are needed in Guam to effectively address the impacts of chronic disease. Guam is home to a diverse population, including CHamorus, the indigenous people of Guam, Filipinos, other Micronesians, and other Asian people. The ethnic diversity in Guam has evolved through the centuries as a result of colonization and ongoing migration, which may be a risk factor linked to the increased prevalence of metabolic disease on the island [25,26]. Migration between and throughout the US-Affiliated Pacific Island (USAPI) region accentuates the burden of disease in USAPI populations. Given the high prevalence of T2DM and OWOB, along with understudied additional risk factors, MetS prevalence in Guam is still unknown.

The Pacific Islands Cohort on Cardiometabolic Health (PICCAH) study was an epidemiological study that collected data on lifestyle factors related to non-communicable diseases (NCDs) from parent–child dyads in Guam, Pohnpei, and Palau to further understand generational associations [27]. The PICCAH study was the first to collect health indicators and lifestyle factors across generations of USAPI populations, an ideal cohort to examine MetS risk factors, which are determined by multiple indicators. To address the research gap, the present study analyzed PICCAH data to examine three aims: first, to determine MetS prevalence among adults and children ages 3–9-year-old in Guam; second, to examine modifiable lifestyle risk factors associated with MetS indicators for children and adults; and third, to evaluate the associations between parent and child MetS risk.

## 2. Materials and Methods

### 2.1. Study Design

This cross-sectional study included data from the PICCAH study. The overall study design and rationale of the PICCAH study have been published elsewhere [27]. This was the first analysis from the PICCAH cohort to assess the relationship of MetS risk factors between children and their biological parents in Guam from 2018 to 2019 only. Parent–child dyads were recruited, including data from 338 adults (21–50 years) and 214 children (3–9 years) in Guam who completed any of the assessments during Visits 1 and 2.

The overall PICCAH study was approved by the University of Guam Committee on Human Research Subjects (CHRS/IRB#19–171 and #20–100). After consent, adult participants completed demographic and lifestyle surveys for themselves and their child during Visit 1. Also during Visit 1, adult and child participants provided consent and assent, respectively, before anthropometric measurements, such as WC (cm), height (cm), weight (kg), and blood pressure (mmHg). Biospecimens (i.e., HDL cholesterol, triglycerides, and fasting blood glucose) from all consented and assented participants were collected during Visit 2. The data collection methods are described in detail in another publication [27].

### 2.2. Study Participants

Included in this analysis were children, 3–9-year-old, related to the PICCAH study’s original aims. Adults who were identified as the primary caregivers of the child participants (N = 338) were grouped by ethnicity, categorized as CHamoru, Filipino, Other Micronesian (i.e., Carolinian, Chuukese, Kiribati, Kosraean, Palauan, Pohnpeian, Yapese, or Marshallese), and Other Race/Ethnicity (e.g., non-Micronesian Pacific Islander, Other Asian, White, or selected more than one race or ethnicity) and years in age as follows: 18–24, 25–34, 35–44, and 45–54.

The total annual household income was reported for all individuals within the same household by the adult participants (parents/caregivers) and used to represent adult and child income categories following previous published cut-off points [8,10]. Only parent’s education level, which was self-reported based on the highest year of schooling completed, was included. The cultural identity of the parent was determined by their ethnic group score (ECS) and U.S. Mainland score (UCS) on a scale used in previous studies with a similar population [26], where the mode of acculturation classified cultural identity as traditional (≤12 ECS and >12 UCS), integrated (12 ≤ ECS and ≤12 UCS), assimilated (>12 ECS and ≤12 UCS), and marginalized (>12 ECS and >12 UCS), as described elsewhere [10,28].

Food insecurity was assessed with one question from the U.S. Department of Agriculture’s Core Food Security Module: “In the past 12 months, how often does money for food run out by the end of the month?” [29]. Household food insecurity was considered present if the respondents chose “Sometimes”, “Most times”, or “Always”, and for “Never” or “Seldom” responses, the household was considered food-secure.

### 2.3. Study Measures

MetS lifestyle factors of interest to this study were gathered from the responses to the parent–child physical activity interaction and sleep questionnaires (parent and child); the International Physical Activity Questionnaire (IPAQ) and Depression, Anxiety, Stress Scales (DASS) tools for sedentary behavior and stress (parents only), respectively; and the screen time questionnaire (child only).

Regarding physical activity (parent and child), physical activity levels were assessed based on the self-reported number of hours (adults) and frequency of activity (child). For adults, weekly physical activity hours were summed and classified as “more active” and “less active” if meeting or not meeting the recommended 2.5 h per week based on the Baecke Physical Activity Questionnaire [30]. Parents reported child activity by responding to the questions “How often does your child do something physically active when he/she has free time?” and “How often does your child participate in organized sports or physical activities with a coach or leader?”, whereby “Never/Almost Never,” “Sometimes,” “Often,” and “Very Often/Always” were coded as 0, 1, 2, and 3 for each question. The responses were combined, and the average of the two responses was categorized as “less active” for a mean between 0 and 2, while the remaining responses were “more active.”

Regarding sedentary behavior (parent), parents’ sedentary behavior was measured by the reported average total hours of sitting activities (e.g., sitting in car, sitting at work, sitting at meals, reading, watching TV) per day. Total daily hours equal to or more than 9 were considered as not meeting the recommendations and defined as “more sedentary” in this study.

Regarding sleep (parent and child), sleep duration was assessed separately by the parent/caregiver as hours asleep at night and during naps for children, hours asleep at night for adults, and whether the number met the recommended sleep duration guidelines for age [31].

Regarding stress (parent), parents’ stress levels were assessed using the self-reported Depression Anxiety Stress Scales (DASS). The Depression Anxiety Stress Scales-21 (DASS-21) was used to assess stress levels [32]. Participants responded to the 7 stress-related items of the scale using a modified response format: “Never (0)”, “Sometimes (1)”, “Often (2)”, and “Almost Always (3)”. The total stress score was calculated by summing the responses to these 7 items. Stress levels were then categorized into two groups: “Less Stressed”, which included participants with normal, mild, or moderate stress scores, and “More Stressed”, which included those with severe or extremely severe stress scores.

Regarding screen time (child only), screen time for children was assessed by the parent-reported average number of hours the child spent on screens during weekdays, categorized as “Never or Rarely” (0–2 h), “Sometimes” (3–5 h), “Often” (6–8 h), or “Very Often/Always” (9–10+ h). A composite score for screen time was calculated by summing weekday and weekend screen time as follows: 0 = Never/Almost Never (0–1 h), 1 = Sometimes (2–4 h), and 2 = Often/Always (5+ h).

Regarding MetS (parent) and MetS risk (child), the IDF criteria for MetS and MetS risk in adults and children, respectively, were applied [6,11] and are outlined in Table 1. For high WC (central obesity) in adults, the IDF uses ethnic-specific values for countries or ethnic groups that do not include Pacific Islanders or the US-Affiliated Pacific region. For this study, waist circumference values for Europids were used, as this category includes populations in the US. Child participants with abdominal obesity were determined to be at risk for MetS with a WC above the 90th percentile using sex- and age-specific cut-off points [11,31]. For children 3, 4, and 5 years of age, the 90th percentile for WC was 55.3 cm, 59.7 cm, and 61.6 cm, respectively, for males, and 54.2 cm, 58.1 cm, and 64.2 cm, respectively, for females [33].

### 2.4. Data Analysis

The MetS Z-scores for adults and children were calculated using formulas by DeBoer and Gurka for non-Hispanic White adults and adolescents, respectively [31,33,34,35]. Demographic data and modifiable lifestyle factors such as sleep, sedentary behavior, physical activity, and stress were evaluated. A chi-square test was used to test for independence to examine the relationship between MetS and each demographic variable. The MetS Z-score was calculated using the following sex-specific formulas for adults and children:

Adults:Females = 7.2591 + 0.0254 × Waist Circumference − 0.0120 × HDL + 0.0075 × SBP + + 0.5800 × ln(Triglycerides) + 0.0203 × GlucoseMales = −5.4559 + 0.0125 × Waist Circumference − 0.0251 × HDL + 0.0047 × SBP + + 0.8244 × ln(Triglycerides) + 0.0106 × Glucose

ChildrenFemales = −4.3757 + 0.4849 × BMI Z-score − 0.0176 × HDL + 0.0257 × SBP + + 0.3172 × ln(Triglycerides) + 0.0083 × GlucoseMales = −4.9310 + 0.2804 × BMI Z-score- 0.02557 × HDL+ 0.0189 * SBP + + 0.6240 × ln(Triglycerides) + 0.0140 × Glucose

The mean values of MetS indicators, including MetS Z-score, waist circumference (cm), triglycerides (mg/dL), HDL cholesterol (mg/dL), and fasting blood glucose (mg/dL), were compared across different lifestyle risk factor groups using independent *t*-tests.

The relationship between adult MetS Z-scores and child MetS Z-scores among PICCAH participants in Guam was evaluated using linear-regression analyses. Both unadjusted and adjusted models were conducted. The adjusted linear regression controlled for potential confounders, including the parent’s education level and income and the child’s age, to provide adjusted estimates for the association. The results included the adjusted regression coefficient (B), 95% CI, *p*-value, and correlation coefficient (R), with statistical significance determined at a *p*-value of less than 0.05.

## 3. Results

The majority of child participants included in the final analysis (N = 214) were male (51.87%), and there were nearly equal participants in age groups 3–6 y (55.14%) and 7–9 y (55.86%). For adult participants (parents), the majority were female (99.22%), CHamoru (52.10%), and Other Micronesian (26.65%). Most parents were in the 25–35 years age group (46.09%) with a household income of <USD 20,000 (30.89%), high school education (49.1%), and integrated cultural identity (62.69%). The CHamoru population had the highest percentage of MetS (47.1%). The relationship between parents’ ethnicity and MetS was significant (*p* < 0.05), as shown in Table 2.

Most adults (92.3%) had abdominal obesity. The prevalence of MetS indicators in parents and children (respectively) was as follows: high serum triglycerides in 18.93% and 2.34% of the sample and high fasting blood glucose in 25.44% and 0.93% of participants. In adults, the prevalence of high blood pressure and low HDL levels was 31.66% and 60.65%, respectively. Using the IDF criteria [31], 39.05% of adults had MetS and 7.69% of children were at risk for MetS, as shown in Table 3.

MetS indicators and MetS Z-scores for both parents and children were examined according to lifestyle risk factors, and no significant differences were found, as shown in Table 4.

Children’s MetS Z-scores were significantly correlated with parents’ MetS Z-scores (Pearson correlation = 0.288, *p* < 0.001). The multiple-regression model showed that child MetS Z-score could be predicted by parent MetS Z-score even after adjusting for the parent’s education level and income and the child’s age (*p* = 0.001). An increase of one unit in the adults’ MetS Z-score was associated with an increase in the child’s MetS Z-score of 0.126 (Table 5).

## 4. Discussion

This study is the first to describe the prevalence of MetS among adults in Guam, an understudied population. The results correlate with previous studies that show that adults in Guam suffer from high rates of T2DM and OWOB [7,14]. Our results show that the prevalence of MetS among adults in Guam (39.05%) was higher than that among adults in the US from 2007 to 2012 (34.2%) [8]. The 2019 IDF criteria for central obesity in adults considers sex- and country-/ethnicity-specific WC cut-off points for Europids, South Asians, Chinese, Japanese, Ethnic South and Central Americans, Sub-Saharan Africans, and Eastern Mediterranean and Middle East (Arab) populations but does not include Pacific Islanders and the US-Affiliated Pacific region [31]. Although the IDF uses lower thresholds of WC for South and East Asian men and women, WC cut-off points of 90 cm for men and 80 cm for women in the Asia–Pacific region have been used, which were the same values used in this study [36].

Guam’s diverse population represents part of the Native Hawai`ian and Other Pacific Islander (NHOPI) and Asian races typically underrepresented in U.S. reports due to aggregated data and geographically isolated states and territories not included in data collection like the NHANES. The NHOPI population is defined, for the purposes of minimum race/ethnicity federal reporting, as individuals with origins in any of the original peoples of Hawai`i, Guam, Samoa, or other Pacific Islands [37]. The NHOPI population is one of the groups at the highest risk for cardiovascular risk factors, with higher prevalences of T2DM, obesity, and cardiovascular disease than other groups in the US [38]. This study disaggregated the NHOPI category and revealed another health disparity related to NCDs aggravating the burden of disease among CHamorus and Other Micronesians and Filipinos living in Guam [39,40]. Known risk factors of MetS include low household income, high school education or lower, and advanced age [38,40,41]. With the exception of age, the adult participants in this study shared similar characteristics, and the findings confirmed the education level to be significantly associated with MetS in adults.

These outcomes add to the body of literature noting that Guam adults are at significant risk of heart disease and other chronic illnesses such as cancer and T2DM, many of which are preventable through early lifestyle modifications [39,41]. More specifically, MetS indicators, such as a high waist circumference, elevated fasting blood glucose, and high blood pressure, can be treated through lifestyle behavior changes, including healthy eating, regular physical activity, and stress management [42,43]. These results warrant future interventions, as recent studies in Guam have demonstrated that adults have poor diets and are sedentary [12,44,45]. One study in Pacific and New Zealand women found an inverse association between high fiber intake and body fat percentage and the odds of metabolic syndrome [46]. The main sources of fiber in the above study population were discretionary fast foods, whole-grain breads, and cereals, aligning with the nutrition transition which has occurred across the USAP [46,47,48,49]. Across the USAP, there are many varieties of high-fiber root crops and starches, like taro, yam, tapioca, breadfruit, and plantains, that could provide a localized source of intervention [50,51].

This study was also the first study to estimate MetS risk in children as young as 3 years of age and overall prevalence in children in the Western Pacific. Among children, the prevalence of abdominal obesity was 7.69%, which was slightly lower than a recent study among 2–8-year-olds in Guam (8.94%) [10]. However, a later publication determined optimal WC cut-off points for children aged 2 to 8 years in the US-Affiliated Pacific (USAP) region, which includes ethnic groups prevalent in Guam [40]. Future studies are critical to determine MetS risk in this population using these relevant cut-off points. Another consideration in the USAP region is cultural identity related to the diverse populations and complex histories characterizing this area. A recent study in Guam found that children from families with an integrated cultural identity were twice as likely to be OWOB, which aligns with the majority of parents in this study (62.7%) reporting having an integrated cultural identity, which may be attributed to MetS risk in this child population [10].

Despite no significant differences in MetS indicators and MetS Z-scores for parents and children by lifestyle risk factors, MetS indicators were lower for parents with less stress compared to parents with more stress for all factors, except HDL. These indicators were also lower for children who met sleep recommendations compared to children who did not for all indicators examined—the expected direction. Additionally, this might have also been related to the fact that cardiovascular risk factors have been reported to cluster within families [20]. In a meta-analysis on the relationship with MetS risk in offsprings of parents with MetS, it was found that the offsprings of at least one parent with MetS had a higher risk of developing MetS. A higher risk was also found in children of mothers with MetS, which is similar to our population [52]. These results provide a basis for lifestyle interventions that involve both parents and children, related to the MetS child–parent linear association found. Although the children in this study, based on data collected in Guam during 2011–2012, exhibited a lower prevalence of abdominal obesity (high waist circumference) than children in the U.S. of a similar age [53], the high MetS prevalence among their parents and exposure to an obesogenic environment warrants early lifestyle interventions for children in Guam [54]. Interventions involving different household members and the support network of a child have been shown to be effective to improving lifestyle behaviors and health outcomes [55,56], as CHamorus and Filipinos alike heavily rely on a large social network to raise children and engage in cultural practices involving food and activity [57,58]. Additionally, multi-sector interventions have been shown to decrease acanthosis nigricans and obesity in young children [59], and studies to capture effects in parents and caregivers are needed. Therefore, multi-component interventions for households with children may be a viable approach to curbing the burden of MetS and other NCDs.

This study has several limitations. First, the small sample size was due to incomplete data, with only a limited number of adults having completed blood work and an even smaller number of children having available laboratory results. Lifestyle risk factors—such as physical activity, sleep, and others listed in Table 4—were self-reported and often missing. For this reason, we did not restrict the analysis to only parent–child dyads with fully completed data, particularly since some analyses were conducted on adults only. The issue of unmatched participants and incomplete data further limited the ability to examine full dyadic relationships. Being the first study to examine MetS risk in early childhood, there is limited literature on whether or not abdominal obesity at these ages predicts MetS or increases risk for MetS [20,60,61]. Additionally, there are no established formulas for calculating MetS Z-scores specifically for Pacific Islander populations or young children. We applied formulas developed by Gurka et al. [33], which have not been validated for this population. Future studies should aim to address this gap. Another limitation of this study is that the lifestyle factors measured did not include all possible modifiable risk factors, like diet, smoking, and consumption of sugar-sweetened beverages and alcohol [62,63,64]. However, the PICCAH study collected these data for later analyses. Furthermore, lifestyle behaviors are complex, and a more comprehensive analysis of lifestyle data is warranted to identify effective intervention strategies [61,65,66]. As this was only the first analysis of the PICCAH dataset, future publications will be able to identify potential lifestyle interventions for this population.

## 5. Conclusions

Child–parent risk for MetS was directly correlated in populations in Guam. The high prevalence of adult MetS in Guam demonstrates a critical need for interventions involving both parents and children. Expanding the analysis to assess the relationship of other lifestyle factors, like diet, among parent–child dyads is necessary to refine such intervention programs.

## Figures and Tables

**Table 1 ijerph-22-00611-t001:** International Diabetes Federation (IDF) criteria for metabolic syndrome (MetS, adults) and MetS risk (children).

IDF Criteria ^1^	MetS (Adult)	At Risk for MetS (Child, 6–<10 y)
Waist Circumference (WC) ^2^	Male: >94 cm	>90th percentile, males 6 y: >68 cm 7 y: >72.5 cm 8 y: >77.4 cm 9 y: >85.3 cm
	Female: >80 cm	>90th percentile, females 6 y: >68 cm 7 y: >73.5 cm 8 y: >78.0 cm 9 y: >77.3 cm
Plus any 2 of these 4 factors:		None established for this age group.
Triglycerides	≥150 mg/dL Or receiving treatment for dyslipidemia	
HDL ^3^ Cholesterol	Males (all): <40 mg/dL Or receiving treatment for dyslipidemia	
	Females (all): <50 mg/dL Or receiving treatment for dyslipidemia	
Blood Pressure	Systolic ≥ 130 mm Hg Diastolic ≥ 85 mm Hg Or receiving treatment for hypertension	
Fasting Blood Glucose	≥100 mg/dL Or diagnosed T2DM	

^1^ International Diabetes Federation criteria (2021); ^2^ adult WC: IDF country/ethnic group values for Europids were used; child WC: 90th percentiles for children and adolescents 2–19 years from the National Center for Health Statistics, National Health and Nutrition Examination Survey 2015–2018; ^3^ HDL: high-density lipoprotein.

**Table 2 ijerph-22-00611-t002:** Characteristics of PICCAH study adult participants with or without metabolic syndrome (MetS).

Adult Characteristics		Total N	MetS ^1^ n (%)	No MetS ^1^ n (%)	*p*-Value *
Age group (years)					0.107
	18–24	13	3 (23.1)	10 (76.9)	
	25–35	118	40 (33.9)	78 (66.1)	
	35–44	100	42 (42.0)	58 (58.0)	
	45–54	25	14 (56.0)	11 (44.0)	
Ethnicity					0.005
	CHamoru	174	82 * (47.1)	92 * (52.9)	
	Filipino	55	13 * (23.6)	42 * (76.4)	
	Other Micronesian	89	28 * (31.5)	61 * (68.5)	
	Other Race/Ethnicity ^2^	10	5 * (50.0)	5 * (50.0)	
Household income (USD)					0.692
	<USD 20,000	80	29 (36.3)	51 (63.8)	
	USD 20,000–USD 34,999	67	30 (44.8)	37 (55.2)	
	USD 35,000–USD 59,999	46	17 (37.0)	29 (63.0)	
	>USD 60,000	66	24 (36.4)	42 (63.6)	
Education level					0.095
	Less than high school	10	6 (60.0)	4 (40.0)	
	High school	164	72 (43.9)	92 (56.1)	
	College (1–3 years)	65	21 (32.3)	64 (67.4)	
	College (4+ years)	95	31 (32.6)	64 (67.4)	
Food-secure ^3^					0.753
	Yes	161	67 (41.6)	94 (58.4)	
	No	153	61 (39.9)	92 (60.1)	

^1^ MetS was defined using the 2021 IDF criteria for adults; ^2^ non-Micronesian, non-CHamoru, and non-Filipino; ^3^ household food security was measured by one question from the US Department of Agriculture’s Core Food Security Module, “In the past 12 months how often did your money for food run out before the end of the month?”, whereby “Yes” represents responses that were “never” or “seldom” and “No” represents “sometimes”, “most times”, or “always” responses; * chi-square tests with significance at *p* < 0.05.

**Table 3 ijerph-22-00611-t003:** Prevalence of metabolic syndrome (MetS) and MetS indicators among adult and child PICCAH study participants.

MetS Indicators ^1^		Adult (Parent)			Child		
		n	%	95% CI	n	%	95% CI
Central Obesity ^2^		312 ^a^	92.31	(89.47, 95.15)	15	7.69	(3.95, 11.43)
Lipid Profile							
	Low HDL	205 ^a^	60.65	(55.44, 65.86)	33 ^d^	15.42	(10.54, 20.30)
	High Triglycerides	64 ^a^	18.93	(14.76, 23.11)	5 ^d^	2.34	(0.30, 4.38)
Glucose							
	Hyperglycemia	86 ^a^	25.44	(20.80, 30.09)	2 ^d^	0.93	(0.00, 2.22)
	High Hemoglobin A1c	50 ^b^	15.02	(11.18, 18.85)	1 ^e^	0.47	(0.00, 1.39)
High Blood Pressure		107 ^a^	31.66	(26.70, 36.62)	1 ^f^	0.47	(0.00, 1.38)
MetS (Adults) ^2^		132 ^a^	39.05	(33.85, 44.25)	-	-	-
At Risk for MetS (Child) ^3^		-	-	-	15 ^c^	7.69	(3.95, 11.43)
Z-Score (Mean)		0.48 ^a^		(0.31, 0.65)	−0.52 ^d^		(−0.62, −0.43)

^1^ International Diabetes Federation criteria for MetS (2021); ^2^ metabolic syndrome (MetS, adults) was determined with central obesity and any 2 of 4 remaining risk factors in the 2021 IDF criteria; ^3^ at risk for MetS was determined with age- and sex-specific waist circumference cut-off points greater than the 90th percentile (abdominal obesity); all child participants were <10 y; ^a^ total adults with data to determine MetS risk factors and MetS, N = 338; ^b^ total adults with data to determine MetS risk factor, N = 333; ^c^ total children with data to determine MetS risk factor, N = 195; ^d^ total children with data to determine MetS risk factor, N = 214; ^e^ total children with data to determine MetS risk factor, N = 211; and ^f^ total children with data to determine MetS risk factor, N = 215.

**Table 4 ijerph-22-00611-t004:** Metabolic syndrome (MetS) indicators (mean ± standard deviation, SD) among adult and child participants according to lifestyle risk factors.

		MetS Indicators				
		MetS Z-Score	Waist Circumference (cm)	Triglycerides (mg/dL)	HDL Cholesterol (mg/dL)	Fasting Blood Glucose (mg/dL)
Adult Behaviors (Mean ± SD)						
Physical Activity ^1^ (h/day)						
	Less active	0.15 ± 1.17	101.24 ± 16.09	107.11 ± 58.35	48.50 ± 11.62	95.21 ± 37.69
	More active	0.54 ± 1.64	103.19 ± 16.48	109.87 ± 76.67	48.61 ± 12.06	107.54 ± 60.89
Sedentary ^2^ (hrs/day)						
	Less sedentary	0.44 ± 1.51	102.33 ± 17.22	111.81 ± 67.91	49.85 ± 12.31	105.09 ± 56.08
	More sedentary	0.27 ± 1.33	102.09 ± 15.62	103.21 ± 59.13	52.56 ± 10.85	97.88 ± 43.64
Sleep						
	<8 h	0.46 ± 1.52	102.27 ± 17.26	108.52 ± 74.71	48.06 ± 11.32	105.21 ± 58.10
	>8 h	0.63 ± 1.74	103.98 ± 16.47	110.02 ± 74.73	48.87 ± 12.67	111.70 ± 64.28
Stress ^3^						
	Less stress	0.44 ± 1.50	102.82 ± 15.98	108.95 ± 70.18	48.75 ± 11.99	104.18 ± 52.95
	More stress	0.87 ± 2.27	103.71 ± 20.80	116.06 ± 109.01	47.94 ± 12.25	123.44 ± 94.91
Child Behaviors (Mean ± SD)						
Physical Activity ^4^						
	Less active	−0.56 ± 0.74	58.25 ± 13.37	68.95 ± 34.89	49.60 ± 11.47	79.26 ± 9.53
	More active	−0.48 ± 0.65	57.90 ± 8.25	63.58 ± 25.85	52.10 ± 12.70	81.58 ± 8.29
Screen Time (hrs/day)						
	<2 h	−0.32 ± 0.76	57.76 ± 10.12	75.36 ± 45.74	47.77 ± 8.30	79.36 ± 11.46
	2+ h	−0.54 ± 0.69	58.15 ± 11.64	66.73 ± 30.06	50.67 ± 12.25	80.14 ± 8.71
Sleep Recommendation ^5^						
	Met	−0.53 ± 0.69	57.64 ± 7.10	66.87 ± 32.26	50.17 ± 10.48	78.43 ± 8.81
	Not met	−0.51 ± 0.71	58.34 ± 13.64	68.37 ± 31.46	50.43 ± 13.04	81.42 ± 9.27

^1^ Adults’ physical activity recommendations are 150 min or 2.5 h per week. Participants reported their activity in hour intervals, and the rounded value of 3.0 h per week was used as the cut-off point; ^2^ adults’ sedentary activity recommendations are less than 9 h per day. Participants reported their sedentary activity in hour intervals; ^3^ adults’ stress cut-off points were based on the Depression and Anxiety Stress Scales (DASS-21) to assess less (normal to mild) and more (moderate, severe, and extremely severe) stress; ^4^ child physical activity was based on the average of two questions with a response based on a Likert scale (i.e., “never” (1), “almost never” (2), “very often” (3), and “always” (4)), whereby average scores of 1–2.9 were “less active” and 3–4 were “more active”. ^5^ The minimum hours recommended for 3–5-year-old and 6–12-year-old children are 10 and 9 h, respectively.

**Table 5 ijerph-22-00611-t005:** Association between adult MetS Z-score and child at-risk MetS Z-score among PICCAH participants.

	Unadjusted Linear Regression				Adjusted Linear Regression *			
	B	95% CI	*p*-Value	R	B	95% CI	*p*-Value	R
Adult MetS Z-score	0.128	(0.068, 0.188)	<0.001	0.288	0.126	(0.065, 0.186)	0.001	0.297

* Adjusted for parent’s education level and income and child’s age.

## Data Availability

Data is available upon request.

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
