# Peer review of "The Association of Cardiometabolic Risk Factors in Parent–Child Dyads in Guam: Pacific Islands Cohort on Cardiometabolic Health Study"

_ijerph, 2025, doi:10.3390/ijerph22040611_

Round 1
Reviewer 1 Report
Comments and Suggestions for Authors Review of the Paper: "The Association of Cardiometabolic Risk Factors in Parent-Child Dyads in Guam: PICCAH Study" 1. Main Question Addressed by the Research The main question addressed by this research is the prevalence of metabolic syndrome (MetS) among adults and children in Guam and the association of MetS risk factors between parent-child dyads. Which is clear 2. Originality and Relevance to the Field The topic is both original and highly relevant to the field of public health and epidemiology in Guam, a region with limited data on this condition, and highlights the intergenerational transmission of MetS risk factors. 3. Contribution to the Subject Area This study adds to the subject area by: Providing the first estimates of MetS prevalence among adults in Guam. Highlighting the high prevalence of MetS and its risk factors in both adults and children. 4. Methodological Improvements The authors could consider the following improvements: Dietary Factors: Including more detailed dietary assessments to better understand the role of diet in MetS risk. 5. Consistency of Conclusions with Evidence and Arguments The conclusions are consistent with the evidence and arguments presented. The study effectively addresses the main question posed by demonstrating the high prevalence of MetS in Guam and the significant association between parent and child MetS risk. The findings support the need for targeted interventions to address MetS in this population. 6. Appropriateness of References The references are appropriate and relevant to the study. They include recent and pertinent studies that support the research findings and provide a solid foundation for the study's conclusions. 7. Additional Comments on Figures Figures: The figures, if any, should visually represent the key findings and trends in the data. They should be clear, labeled accurately, and complement the information presented in the tables. The tables needs to be more organizedAuthor Response
Comment 1: Main Question Addressed by the Research The main question addressed by this research is the prevalence of metabolic syndrome (MetS) among adults and children in Guam and the association of MetS risk factors between parent-child dyads. Which is clear
Response 1: Thank you.
Comment 2: Originality and Relevance to the Field The topic is both original and highly relevant to the field of public health and epidemiology in Guam, a region with limited data on this condition, and highlights the intergenerational transmission of MetS risk factors.
Response 2: We appreciate that this work is valued.
Comment 3: Contribution to the Subject Area This study adds to the subject area by: Providing the first estimates of MetS prevalence among adults in Guam. Highlighting the high prevalence of MetS and its risk factors in both adults and children.
Response 3: This population is underrepresented in research and we are excited to be a part of highlighting this.
Comment 4: Methodological Improvements The authors could consider the following improvements: Dietary Factors: Including more detailed dietary assessments to better understand the role of diet in MetS risk.
Response 4: Although diet was collected and methods were described in the PICCAH Cohort's first publication (Ref 27 in the paper), we decided not to include as we are currently working on analysis for another manuscript. When diet analysis is completed, we will have that paper followed by more complex analysis evaluating association with other lifestyle factors and disease- to expand this work. We pointed to this in the Abstract (Lines 28-29), Discussion (281-282), and Conclusion (349-350)
Comment 5: Consistency of Conclusions with Evidence and Arguments The conclusions are consistent with the evidence and arguments presented. The study effectively addresses the main question posed by demonstrating the high prevalence of MetS in Guam and the significant association between parent and child MetS risk. The findings support the need for targeted interventions to address MetS in this population.
Response 5: We agree.
Comment 6: Appropriateness of References The references are appropriate and relevant to the study. They include recent and pertinent studies that support the research findings and provide a solid foundation for the study's conclusions.
Response 6: Thank you.
Comment 7: Additional Comments on Figures Figures: The figures, if any, should visually represent the key findings and trends in the data. They should be clear, labeled accurately, and complement the information presented in the tables. The tables needs to be more organized.
Response 7: We have organized and clarified labeling in all tables as follows:
- Table 1: changed PICCAH variable to IDF criteria in headers and updated footnotes to clarify parameters
- Table 2: Units were added to adult characteristics, where applicable. Revised footnote numbers to be consistent (i.e., all numerical). Reformatted.
- Table 3: Simplified Title. Reformatted by removing N (total) column and added N’s to footnote.
- Table 4: Revised title and headers.
- Table 5: Revised title.
Reviewer 2 Report
Comments and Suggestions for Authors The authors reported the MetS prevalence, lifestyle risk factors associated with MetS indicators among adults and 3-9-year-old children, and relationship between adult and children MetS Z-scores in Guam, a specific population which have had limited studied. This research expands the knowledge of metabolism related diseases in this specific population. In methods, there are 338 adults (21-50 years) and 214 children (3-9 years). Inadequate numbers of adults and children suggest that they are not truly child-parent dyad which been studied. It is suggested that the authors modify the title of this manuscript. In Table 2, the sum of age group is 256 (13 + 118 + 100+ 25), the sum of ethnicity group is 328 (174 + 55 + 89 + 10). These numbers are less than 338 adult subjects. Same errors in income group, cultural identity and food secure groups. The authors missed the number in education level group. In Results, the authors stated that “The prevalence of MetS indicators 220 in parents and children (respectively) were high serum triglycerides (18.5% and 2.7%) 221 and high fasting blood glucose (25% and 1.8%)”. This data did not match the data in Table 3. Please carefully check other data in Results and Tables. Most of them did not match. The references are appropriate in this manuscript.Author Response
Comment 1: Inadequate numbers of adults and children suggest that they are not truly child-parent dyad which been studied. It is suggested that the authors modify the title of this manuscript. In Table 2, the sum of age group is 256 (13 + 118 + 100+ 25), the sum of ethnicity group is 328 (174 + 55 + 89 + 10). These numbers are less than 338 adult subjects. Same errors in income group, cultural identity and food secure groups.
Response 1: The overall PICCAH study recruited parent-child dyads intentionally as part of the study design (Ref 27); therefore, dyads were included in this study. Due to the nature of the community-based study, not all participants were able to complete all assessments, hence the different numbers for study variables for both parents and children. Considering the importance of this study to the field and population, we included any and data collected. Lines 116-118 was revised, "Parent-child dyads were recruited and data included were from 338 adults (21-50 years) and 214 children (3-9 years) in Guam who completed any of the assessments at Visit 1 and 2." We revised the limitations in the Discussion (Lines 325-332, copied below) to describe our approach to the data analysis in this study.
"First, the small sample size was due to incomplete data, with only a limited number of adults having completed blood work and an even smaller number of children with available laboratory results. Lifestyle risk factors – such as physical activity, sleep, and others listed in Table 4 – were self-reported and often missing. For this reason, we did not restrict the analysis to only parent-child dyads with fully completed data, particularly since some analyses were conducted on adults only. The issue of unmatched participants and incomplete data further limited the ability to examine full dyadic relationships."
Comment 2: In Table 2, The authors missed the number in education level group.
Response 2: Thank you for raising this to our attention. We have now entered all values in Table 2 for Education Level.
Comment 3: In Results, the authors stated that “The prevalence of MetS indicators 220 in parents and children (respectively) were high serum triglycerides (18.5% and 2.7%) 221 and high fasting blood glucose (25% and 1.8%)”. This data did not match the data in Table 3. Please carefully check other data in Results and Tables. Most of them did not match.
Response 3: The Results section was updated to align with Table 2. Thank you.
Reviewer 3 Report
Comments and Suggestions for Authors
The study provides a valuable investigation into the association of cardiometabolic risk factors in parent-child dyads in Guam, utilizing data from the Pacific Islands Cohort on Cardiometabolic Health (PICCAH) study. Some comments follow:
The research gap should be more explicitly defined to strengthen the rationale for conducting this study. Additionally, a brief discussion on why the PICCAH study sample is particularly suitable for examining MetS risk in Guam would help establish the significance of the study’s approach. Providing more details on the recruitment of participants and the inclusion/exclusion criteria would further improve the study’s transparency.
The methodology is detailed, but certain aspects require further clarification. The study effectively assesses lifestyle factors such as physical activity, sedentary behavior, and sleep; however, dietary intake and other modifiable risk factors such as smoking and alcohol consumption were not considered. If these variables were collected but not included in the analysis, it would be helpful to mention them as potential factors for future research. Furthermore, the study uses International Diabetes Federation (IDF) criteria to define MetS, but Pacific Islanders are not explicitly included in their recommendations. Justifying the use of Europid values for waist circumference cut-offs is necessary to ensure accuracy in defining MetS for this population. In addition, more details on how the MetS Z-score was calculated and how confounders were adjusted in the statistical models would strengthen the validity of the results.
The results section is well-organized, but some figures require clearer labeling, and statistical significance should be consistently reported, including p-values and confidence intervals. The study finds that MetS risk in children is relatively low (7.69%) compared to adults (39.05%), raising questions about whether this is due to early-stage development or protective factors in childhood. Further discussion of this discrepancy would enhance the interpretation of findings. Additionally, while the manuscript mentions the role of socioeconomic and cultural factors such as income, education, and cultural identity, a more in-depth discussion on how these variables influence MetS prevalence in Guam would provide a broader perspective.
The discussion section is comprehensive and well-structured, but it would benefit from comparisons with other studies on MetS in Pacific Island populations or other minority groups in the U.S. The finding that parent and child MetS risk are directly correlated supports the need for family-based interventions. However, an exploration of the relative contributions of genetic and environmental influences would provide deeper insight into the mechanisms driving this association. The study examines stress and sleep as lifestyle factors, but neither showed a significant connection to MetS risk. A discussion of potential reasons for this, such as the small sample size or limitations of self-reported data, would add context. Additionally, the study briefly discusses obesogenic environments in Guam but could expand on how dietary patterns, physical activity infrastructure, and public health policies contribute to MetS risk in this population.
The limitations of the study should be clearly acknowledged, particularly regarding the relatively small sample size, which may have impacted the statistical power of the findings. The study does not directly measure dietary habits, which is a crucial factor in MetS development. Future research should incorporate dietary assessments to provide a more comprehensive analysis of risk factors. Additionally, exploring longitudinal trends in MetS progression among children would offer valuable insights into how risk factors evolve over time.
Comments on the Quality of English Language
The English could be improved to more clearly express the research.
Author Response
Comment 1: The research gap should be more explicitly defined to strengthen the rationale for conducting this study. Additionally, a brief discussion on why the PICCAH study sample is particularly suitable for examining MetS risk in Guam would help establish the significance of the study’s approach. Providing more details on the recruitment of participants and the inclusion/exclusion criteria would further improve the study’s transparency.
Response 1: The last paragraph of the Background was revised (refer to Lines 100-106) and the Discussion was also enhanced to highlight the research gaps (refer to Lines 250-260, 265-271). The PICCAH study was introduced in the background, rather than in Study Design. Refer to lines 96-106. The details of recruitment are described in another publication (Ref 27).
Comment 2:
The methodology is detailed, but certain aspects require further clarification. The study effectively assesses lifestyle factors such as physical activity, sedentary behavior, and sleep; however, dietary intake and other modifiable risk factors such as smoking and alcohol consumption were not considered. If these variables were collected but not included in the analysis, it would be helpful to mention them as potential factors for future research. Furthermore, the study uses International Diabetes Federation (IDF) criteria to define MetS, but Pacific Islanders are not explicitly included in their recommendations. Justifying the use of Europid values for waist circumference cut-offs is necessary to ensure accuracy in defining MetS for this population. In addition, more details on how the MetS Z-score was calculated and how confounders were adjusted in the statistical models would strengthen the validity of the results.
Response 2: Although diet and other lifestyle factors were collected and methods were described in the PICCAH Cohort's first publication (Ref 27 in the paper), we decided not to include as we are currently working on analysis for another manuscript, specifically diet. When diet analysis is completed, we will have that paper followed by more complex analysis evaluating association with other lifestyle factors and disease- to expand this work. We pointed to this in the Abstract (Lines 28-29), Discussion (281-282), and Conclusion (349-350). We further explain the WC cut-points in the Discussion for both adults and children in Lines 254-260 and Lines 293-296, respectively. MetS Z-scores were included in the Methods 2.4 Data Analysis section in Lines 205-216. Thank you for the keen review to strengthen this paper.
Comment 3: The results section is well-organized, but some figures require clearer labeling, and statistical significance should be consistently reported, including p-values and confidence intervals. The study finds that MetS risk in children is relatively low (7.69%) compared to adults (39.05%), raising questions about whether this is due to early-stage development or protective factors in childhood. Further discussion of this discrepancy would enhance the interpretation of findings. Additionally, while the manuscript mentions the role of socioeconomic and cultural factors such as income, education, and cultural identity, a more in-depth discussion on how these variables influence MetS prevalence in Guam would provide a broader perspective.
Response 3: We further explain the WC cut-points in the Discussion for children in Lines 293-296. Revisions were made throughout the Discussion to address these comments overall.
Comment 4: The discussion section is comprehensive and well-structured, but it would benefit from comparisons with other studies on MetS in Pacific Island populations or other minority groups in the U.S. The finding that parent and child MetS risk are directly correlated supports the need for family-based interventions. However, an exploration of the relative contributions of genetic and environmental influences would provide deeper insight into the mechanisms driving this association. The study examines stress and sleep as lifestyle factors, but neither showed a significant connection to MetS risk. A discussion of potential reasons for this, such as the small sample size or limitations of self-reported data, would add context. Additionally, the study briefly discusses obesogenic environments in Guam but could expand on how dietary patterns, physical activity infrastructure, and public health policies contribute to MetS risk in this population.
Response 4: We added References 38, 39, and 47-53 and revised the Discussion section accordingly.
Comment 5: The limitations of the study should be clearly acknowledged, particularly regarding the relatively small sample size, which may have impacted the statistical power of the findings. The study does not directly measure dietary habits, which is a crucial factor in MetS development. Future research should incorporate dietary assessments to provide a more comprehensive analysis of risk factors. Additionally, exploring longitudinal trends in MetS progression among children would offer valuable insights into how risk factors evolve over time.
Response 5: Limitations were revised in the Discussion Lines 325-340. Although diet was collected and methods were described in the PICCAH Cohort's first publication (Ref 27 in the paper), we decided not to include as we are currently working on analysis for another manuscript. When diet analysis is completed, we will have that paper followed by more complex analysis evaluating association with other lifestyle factors and disease- to expand this work. We pointed to this in the Abstract (Lines 28-29), Discussion (281-282), and Conclusion (349-350).
Thank you.
Reviewer 4 Report
Comments and Suggestions for Authors
The current review is a potentially interesting topic to cover and it is a very well written review. However, I suggest following corrections to be made by authors for further improvement.
Major correction:
Manuscripts reference list is completely missing of the references no. 38-54. It should be revised and added in the final version.
Review can benefit from a more precise abstract highlighting the novelty and the relevance of the study
Author's should cite the below related published article with clearly articulating what sets this study apart from previous work and its impact on strengthening the overall contribution
https://pubmed.ncbi.nlm.nih.gov/35897002/
Author Response
Comment 1: Manuscripts reference list is completely missing of the references no. 38-54. It should be revised and added in the final version.
Response 1: Great catch of this oversight. We have added missing references and updated all references.
Comment 2: Review can benefit from a more precise abstract highlighting the novelty and the relevance of the study
Response 2: We have revised the last paragraph of the Background (refer to Lines 100-106) as well as the Discussion to highlight the existing research gaps (refer to Lines 250-260, 265-271). We added References 38, 39, and 47-53 and revised the Discussion section accordingly.
Comment 3: Author's should cite the below related published article with clearly articulating what sets this study apart from previous work and its impact on strengthening the overall contribution
Response 3: This was addressed by moving introduction of PICCAH from Study Design to Background. Additionally, revising the first paragraph of the Methods to emphasize that "This is the first analysis from the PICCAH cohort..." (Lines 114-115) and Ref 27 is only the study rationale and methods.
Round 2
Reviewer 2 Report
Comments and Suggestions for Authors
The authors addressed most of the questions. However, the data which they described in the second paragraph of Results were still not consistent with the data in Table 3. There are two “High Blood Pressure” categories in Table 3 now.
Author Response
Comment 1: The authors addressed most of the questions. However, the data which they described in the second paragraph of Results were still not consistent with the data in Table 3. There are two “High Blood Pressure” categories in Table 3 now.
Response 1: Thank you for the keen review. Paragraph 2 in Results now reflects the values in Table 3. We may have overlooked the updates we made to the Paragraph 2 in Results to match the numbers in Table 3 when merging the versions from contributing authors. The first mention of "high blood pressure" now reads "High Hemoglobin A1c," which was mistakenly kept when remaking the table to ensure proper formatting. While reviewing Table 3's values we also updated the 95% CI values that were overlooked or incorrectly entered when remaking Table 3 for the resubmission.